# Cysteinyl leukotriene receptor 1 is dispensable for osteoclast differentiation and bone resorption

**Hirofumi Fujita**[1]*, **Aoi Ando**[2], **Yohei Mizusawa**[2], **Mitsuaki Ono**[3], **Takako Hattori**[4], **Munenori Habuta**[5], **Toshitaka Oohashi**[3], **Satoshi Kubota**[4], **Hideyo Ohuchi**[1]*

1 Department of Cytology and Histology, Okayama University Faculty of Medicine, Dentistry and Pharmaceutical Sciences, Okayama, Japan, 2 Faculty of Medicine, Okayama University Medical School, Okayama, Japan, 3 Department of Molecular Biology and Biochemistry, Okayama University Faculty of Medicine, Dentistry and Pharmaceutical Sciences, Okayama, Japan, 4 Department of Biochemistry and Molecular Dentistry, Okayama University Faculty of Medicine, Dentistry and Pharmaceutical Sciences, Okayama, Japan, 5 Department of Cytology and Histology, Okayama University Graduate School of Medicine, Dentistry and Pharmaceutical Sciences, Okayama, Japan

* fujita00@md.okayama-u.ac.jp (HF); ohuchi-hideyo@okayama-u.ac.jp (HO)

**Data Availability Statement:** All relevant data are within the manuscript and its Supporting Information files.

## Abstract

Cysteinyl leukotriene receptor 1 (CysLTR1) is a G protein-coupled receptor for the inflammatory lipid mediators cysteinyl leukotrienes, which are involved in smooth muscle constriction, vascular permeability, and macrophage chemokine release. The *Cysltr1* gene encoding CysLTR1 is expressed in the macrophage lineage, including osteoclasts, and the CysLTR1 antagonist Montelukast has been shown to suppress the formation of osteoclasts. However, it currently remains unclear whether CysLTR1 is involved in osteoclast differentiation and bone loss. Therefore, to clarify the role of CysLTR1 in osteoclastogenesis and pathological bone loss, we herein generated CysLTR1 loss-of-function mutant mice by disrupting the *cysltr1* gene using the CRISPR-Cas9 system. These mutant mice had a frameshift mutation resulting in a premature stop codon (*Cysltr1* KO) or an in-frame mutation causing the deletion of the first extracellular loop (*Cysltr1*$^{\Delta 105}$). Bone marrow macrophages (BMM) from these mutant mice lost the intracellular flux of calcium in response to leukotriene D$_4$, indicating that these mutants completely lost the activity of CysLTR1 without triggering genetic compensation. However, disruption of the *Cysltr1* gene did not suppress the formation of osteoclasts from BMM *in vitro*. We also demonstrated that the CysLTR1 antagonist Montelukast suppressed the formation of osteoclasts without functional CysLTR1. On the other hand, disruption of the *Cysltr1* gene partially suppressed the formation of osteoclasts stimulated by leukotriene D$_4$ and did not inhibit that by glutathione, functioning as a substrate in the synthesis of cysteinyl leukotrienes. Disruption of the *Cysltr1* gene did not affect ovariectomy-induced osteoporosis or lipopolysaccharide-induced bone resorption. Collectively, these results suggest that the CysLT-CysLTR1 axis is dispensable for osteoclast differentiation *in vitro* and pathological bone loss, while the leukotriene D$_4$-CysTR1 axis is sufficient to stimulate osteoclast formation. We concluded that the effects of glutathione and Montelukast on osteoclast formation were independent of CysLTR1.

**Funding:** This study was funded by Japan Society for the Promotion of Science (https://www.jsps.go.jp/english/index.html), (JSPS KAKENHI 15K10475, 19K09625, awarded to HF) and the Promotion of Science and Technology in Okayama Prefecture by the MEXT (https://www.mext.go.jp/) (awarded to HO). The funders had no role in study design, data collection and analysis, decision to publish, or preparation of the manuscript.

## Introduction

Osteoclasts, bone resorbing cells, and osteoblasts, bone forming cells, maintain a balance in bone metabolism under physiological conditions [1]. However, under pathological conditions, osteoclasts play a critical role in bone loss in osteoporosis and inflammatory bone destruction [2]. The differentiation and activation of osteoclasts are mediated by the receptor activator of NF-κB (RANK)-RANK ligand (RANKL) signaling pathway. Some inflammatory mediators and their receptors enhance the RANK-RANKL signaling pathway, causing the excessive formation of osteoclasts and bone resorption [3]. However, the underlying molecular mechanisms remain unclear. Therefore, further studies that clarify these mechanisms will contribute to the attenuation of excessive bone loss in bone diseases.

Cysteinyl leukotrienes (CysLTs), including leukotriene C4 ($LTC_4$), $LTD_4$, and $LTE_4$ are metabolites of arachidonic acid and potent lipid mediators of inflammation with various pathophysiological activities [4]. $LTC_4$ is synthesized by $LTC_4$ synthase using $LTA_4$ and reduced glutathione (GSH) as substrates [5], and $LTC_4$ is then exported to the extracellular space [6]. $LTC_4$ is converted to $LTD_4$ by the extracellular enzyme, γ-glutamyl transpeptidase, and $LTD_4$ is subsequently converted to $LTE_4$ by the membrane-bound dipeptidase. These CysLT activities are mediated by the leukotriene receptors CysLTR1 and CysLTR2, and induce bronchial smooth muscle contraction, increased vascular permeability, eosinophil mobilization, macrophage chemokine release, and the secretion of the extracellular matrix protease, MMP [4, 7, 8]. Previous studies showed that CysLTs stimulate isolated avian osteoclasts to form resorption pits on calcified matrices [9] and stimulate bone resorption in organ culture of mouse calvariae [10, 11]. In addition, $LTD_4$ stimulates osteoclast differentiation from a mouse macrophage cell line RAW264.7 in the presence of RANKL [12]. Previous our study showed that the CysLT synthesis substrate GSH stimulated osteoclast differentiation [13]. These findings suggest that CysLT might be a stimulating factor of osteoclast differentiation and bone resorption. On the other hand, the *Cysltr1* gene encoding the CysLTR1 protein was expressed in macrophages and osteoclasts, and also that CysLTR1 antagonists and *Cysltr1* RNAi inhibited osteoclastogenesis and bone loss [8, 14]. However, it currently remains unclear whether CysLT-CysLTR1 signaling pathway is involved in osteoclast differentiation and bone loss because there is currently no information on the role of CysLTR1 in osteoclast differentiation and activity using mice with the complete loss of CysLTR1 activities.

Therefore, we speculated that the CysLT-CysLTR1 signaling pathway is involved in osteoclast differentiation in bone disease with bone loss. In the present study, we investigated the role of CysLTR1 in osteoclast differentiation and pathological bone loss using two *Cysltr1* loss-of-function mutant mice generated by the CRISPR-Cas9 system.

## Materials and methods

### Animals

All animal procedures were approved by the Okayama University Institutional Animal Care and Use Committee (approval number: OKU2020082, OKU2020085, OKU2020585, OKU2020605, and OKU2020606). Mice were obtained from Japan SLC, Inc. (Shizuoka, Japan). All efforts were made to minimize animal suffering.

### Generation of *Cysltr1* loss-of-function mutant mice

Target sequences within the mouse *Cysltr1* gene for single guide RNA (sgRNA) were designed using CRISPRdirect software (https://crispr.dbcls.jp/) and DNA sequence information for *Cysltr1* was obtained from the Ensembl database (http://asia.ensembl.org/index.html, ID:

ENSMUSG00000052821). Two highly specific target sequences in exon 4 were selected. *Plcl1*, *Pde10a*, *Mcc*, and *Ppargc1b* were listed as candidate off-target genes.

The Cas9 protein was obtained from Integrated DNA Technologies, Inc. (Coralville, IA). Two pairs of oligos for the targeting sites of *Cysltr1* were annealed and inserted into the BbsI site of the pSpCas9(BB)-2A-GFP (PX458; Addgene, MA) vector [15]. The sequences of the oligos were as follows: target 1 (5'-CACCGTACACAGAGTAGATCTGCTA-3' and 5'-AAACTA GCAGATCTACTCTGTGTAC-3') and target 2 (5'- CACCACAATAGAGGTTAACGTACA-3' and 5'-AAACTGTACGTTAACCTCTATTGT-3'). Regarding the template synthesis of sgRNAs, PCR was performed with PrimeStar Max polymerase (Takara Bio, Inc., Shiga, Japan), forward primers adding a T7 promoter sequence (TTAATACGACTCACTATAGGGTACACAGA GTAGATCTGCTAGTTTTAG for target 1; TTAATACGACTCACTATAGGGACAATAGAGGTTA ACGTACAGTTTTAG for target 2), a reverse primer (AAAAGCACCGACTCGGTG for targets 1 and 2), and target sequence-inserted PX458 vectors as templates. PCR products were purified with MagExtractor PCR & Gel Clean up and sgRNA *in vitro* transcription was performed using the MEGAshortscript T7 Transcription Kit (Thermo Fisher Scientific, MA). sgRNAs were purified with the MEGAClear Transcription Clean-Up Kit, concentrated by ethanol precipitation, and then dissolved in Opti-MEM I Reduced Serum Medium (Thermo Fisher Scientific). These sgRNAs and the Cas9 protein were mixed and incubated at room temperature for 10 min to form the Cas9/sgRNA complex (Fig 1A).

*In vitro* fertilization was performed with a standard protocol using the BDF1 and C57BL/6 strains [16, 17]. Fertilized eggs were collected and washed 3 times with Opti-MEM and then placed in the gap of electrodes (LF501PT1-10; BEX, Tokyo, Japan), which were filled with Opti-MEM I containing 1 μg/ml Cas9, 100 ng/ml sgRNA for target 1, and 100 ng/ml sgRNA for target 2 (in a total volume of 5 μl). Electroporation was performed under 30 V (3 msec ON + 97 msec OFF) × 6 (±) using an electroporator (NEPA21; Nepa Gene, Ichikawa, Japan). Electroporated eggs were immediately washed 3 times with KSOM (ARK Resource, Kumamoto, Japan) and cultured in KSOM at 37°C in an incubator supplied with 5% of $CO_2$ until the two-cell stage.

Two-cell stage zygotes were collected and transferred to the oviducts of pseudo-pregnant ICR female mice. The litter were obtained, and genomic DNA was extracted from the tail tips of pups and subjected to a PCR analysis of on-target and off-target loci with specific primers (S1 Table) for genotyping. Heterozygous mice were maintained and crossed using standard procedures as an inbred strain.

## Bone marrow macrophages and osteoclast differentiation

Bone marrow cells were harvested from the femurs and tibias of 6- to 20-week-old WT, *Cysltr1* knockout (KO), or *Cysltr1*$^{\Delta105}$ mice as previously reported [13]. Briefly, cells were cultured on a Petri dish in 10 ng/ml M-CSF containing αMEM supplemented with 10% FBS and antibiotics for 2 or 3 days and adhered bone marrow macrophages (BMM) were then harvested for various experiments. Regarding osteoclast differentiation, BMM were plated on 48-well plates and cultured for 3 days in the presence of 10 ng/ml M-CSF and the indicated concentration of RANKL. $LTD_4$ (Cayman Chemical, MI) was diluted with culture medium as a stock solution (50 μM) and stored at -20°C. $LTD_4$ (1 μM) was added to the medium during osteoclast differentiation 1 day after the RANKL stimulation.

## Quantitative reverse transcription PCR (RT-qPCR)

RT-qPCR was performed as previously reported [16]. Total RNA was purified with the RNeasy Mini Kit (Qiagen, CA) from BMM and osteoclasts and then reverse transcription was

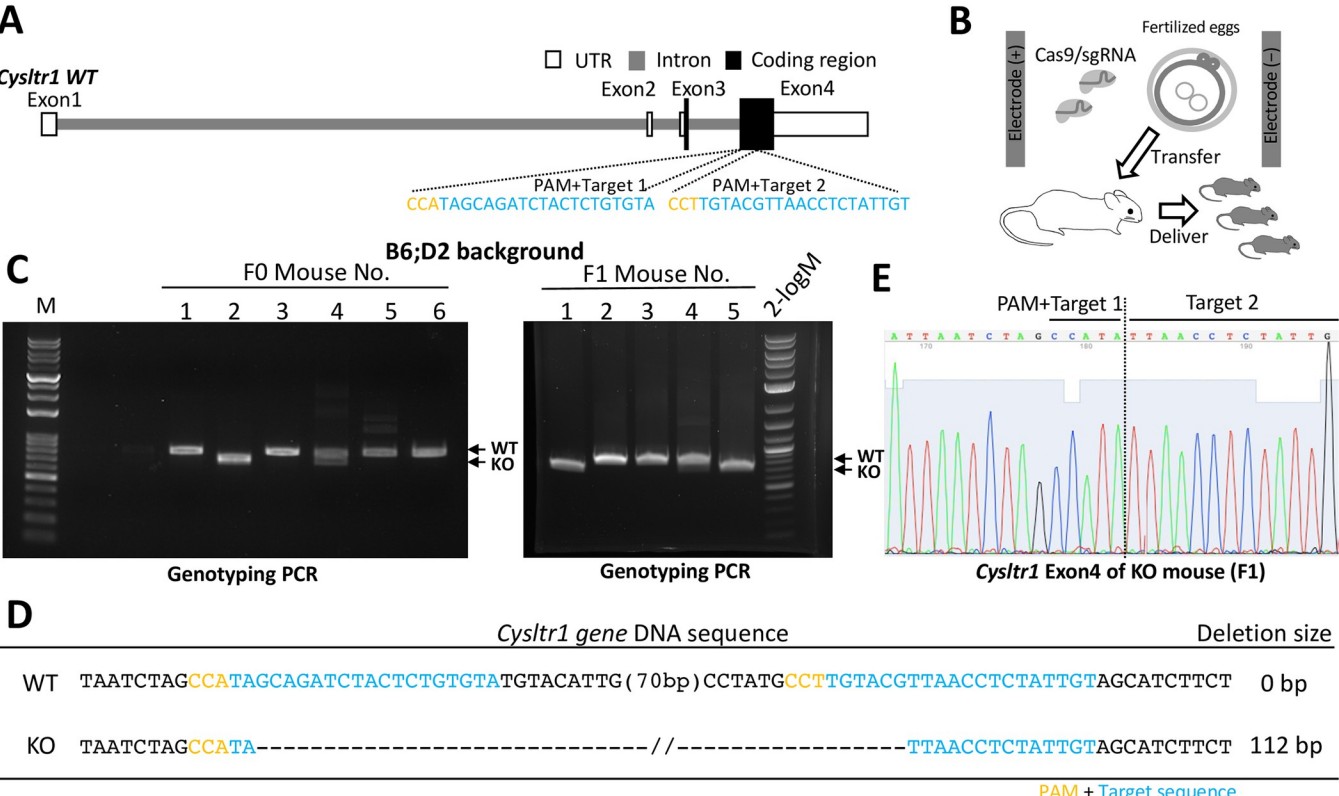

**Fig 1. Cysltr1 gene knockout by the CRISPR-Cas9 system.** (A) Schematic representation of the mouse *Cysltr1 gene* structure. Two sgRNA targeting sites were designed in exon 4. "PAM + Target 1 or 2" is a protospacer adjacent motif plus target sequences for the Cas9/sgRNA complex. (B) Schematic diagram of *Cysltr1* knockout (KO) mouse generation using electroporation to transfer the Cas9/sgRNA complex into one-cell fertilized eggs from BDF1 mice. (C) Genotyping PCR of F0 (left) and F1 (right) pups after *Cysltr1* gene editing in BDF1. The *Cysltr1* wild-type (WT) amplicon size is 818 bp and the KO amplicon size is 706 bp. The F0 female mouse #4 was mated to a WT male mouse, providing the KO strain (F1 mice). (D, E) DNA sequences after *Cysltr1* gene editing. (D) DNA sequences of target sites from WT and KO mice. The exact deletion size in the KO mouse was 112 bp. "-" indicates deleted nucleotide bases. (E) Sequencing chromatogram.

conducted with the ReverTra Ace qPCR RT Master Mix (TOYOBO, Osaka, Japan). qPCR was performed using the LightCycler Nano Instrument and FastStart Essential DNA Green Master Mix. Primer sets used are shown in S1 Table. The *Cysltr1* primer set for qPCR may only amplify the WT *Cysltr1* gene because the forward primer was designed in the deleted DNA sequence of *Cysltr1*.

## Calcium mobilization

Intracellular calcium mobilization was analyzed with Calcium Kit-Fura 2 (Dojindo, Kumamoto, Japan) and the high throughput plate reader FlexStation 3 (Molecular Devices, CA) according to the manufacturer's instructions. BMM ($2 \times 10^5$ cells/ml, 100 μl/well) were seeded on a 96-well tissue culture plate and cultured overnight. Cells were incubated with 5 μg/ml Fura-2-AM at 37°C for 30 min in the presence of 0.04% Pluronic F127 and 1.25 mM Probenecid. Cells were stimulated with 1 μM $LTD_4$ and 50 μM ATP and Fura-2 fluorescence was measured at an emission of 510 nm after excitation at 340 nm ($Ca^{2+}$-bound dye) and 380 nm ($Ca^{2+}$-unbound dye) (time 900 sec; interval 5 sec). Intracellular-free $Ca^{2+}$ concentrations were calculated by the fluorescence ratio at an emission wavelength of 510 nm in response to excitation at 340/380 nm.

## Tartrate-resistant acid phosphatase (TRAP) staining

Eight-week-old WT or *Cysltr1* KO mice were anesthetized with isoflurane, sacrificed, and femurs were harvested and fixed with 4% (w/v) paraformaldehyde at 4˚C for 48 h. The femur was decalcified with 10% (w/v) EDTA solution at room temperature for 14 days. Following decalcification, bone tissues were trimmed off, and specimens were dehydrated with graded ethanol and embedded in paraffin. Sections of these tissues cut at a thickness of 5 μm were stained with TRAP staining solution (50 mM sodium tartrate and 45 mM sodium acetate, pH 5.0, 0.1 mg/ml naphthol AS-MX phosphate, and 0.6 mg/ml Fast red violet LB salt) for 15 min. Nuclei were stained with Mayer's hematoxylin. Number of TRAP positive cells per 1 mm of bone perimeter (Oc. N/B.Pm) and percentage of osteoclast surface in femur bone surface (Oc. S/BS) were quantified using NIH ImageJ software.

After osteoclast differentiation *in vitro*, cells were fixed using 10% (v/v) formalin for 10 min, washed with distilled water, incubated with TRAP staining solution for 15 min, and then washed with PBS. The number of osteoclasts (TRAP-positive multinucleated cells with more than three nuclei) per well (0.77 cm$^2$) were counted under a microscope.

## Pit formation assay

The bone-resorbing activity of osteoclasts was assayed using the Osteo-Assay Plate as previously reported [13]. BMMs were plated on a 96-well Osteo-Assay Plate and cultured with 10 ng/ml M-CSF, 100 ng/ ml RANKL. Medium supplemented with M-CSF and RANKL was changed every 2 days. After six days of RANKL stimulation, cells were removed by an incubation with $NH_4OH$. The bottom of the plate was stained with calcein and washed with distilled water five times. The resorption pit area was visualized with a fluorescence microscope and measured using ImageJ software (National Institutes of Health, Bethesda, MD).

## Ovariectomy-induced osteoporosis

*Cysltr1* heterozygous mice as control and KO female mice (10-week-old, SPF, group feeding) underwent bilateral ovariectomy or a sham operation in which the bilateral ovaries were exteriorized, but not removed, under anesthesia by mixed anesthetic agents (0.375 mg/kg medetomidine hydrochloride, 4 mg/kg midazolam, and 5 mg/kg butorphanol) [18, 19]. Mice were sacrificed 4 weeks after the surgical procedure and the femur and uterus were harvested and fixed with 4% paraformaldehyde. All specimens were scanned using micro-computed tomography (micro-CT; Skyscan1174, Bruker, Aartselaar, Belgium). Scans were performed at a resolution of 64 μm. 3D bone images were reconstructed from slice data and bone volume per total volume (BV/TV), trabecular number (Tb.N), trabecular separation (Tb.Sp), and trabecular thickness (Tb.Th) were analyzed using SkyScan software (Nrecon, CTAn, CTvol, and Ctvox, SkyScan) [20]. The region of interest of trabecular bone was a width of 1.0 mm located 0.5 mm from the growth plate.

## CT analysis of inflammatory bone erosion

Inflammatory bone erosion mouse models were made as previously described [16]. The bregma of WT and *Cysltr1$^{Δ105}$* male mice (6 weeks old, SPF, group feeding) was subcutaneously injected with LPS (12,5 mg/kg). On day 5, mice were sacrificed, and calvarial bones were harvested and fixed with 4% paraformaldehyde for 48 h. A CT analysis was performed with Latheeta-200 (Hitachi Aloka Medical, Tokyo, Japan). 3D bone images were constructed from slice data by a volume-rendering method using VGstudio max 2.1 software (Nihon Visual Science, Tokyo, Japan). Osteolytic lesion areas were measured using NIH ImageJ software.

### Statistical analysis

Statistical analyses were performed using an unpaired two-tailed Student's *t*-test for comparisons between two groups, and a one-way analysis of variance (ANOVA) followed by Tukey's honestly significant difference test for experiments involving more than two groups. Results from *in vivo* data and qPCR data are expressed as the mean ± SE, and *in vitro* results as the mean ± SD. *p* values <0.05 were considered to indicate significant differences. Data were obtained from biological (*in vivo* experiments) or technical (*in vitro* experiments) replicates as shown by the number of samples examined in each legend.

## Results

### Generation of *Cysltr1* mutant mice by the CRISPR-Cas9 system

We initially performed an *in silico* analysis of *Cysltr1* gene expression in the osteoclast lineage and found that *Cysltr1* was highly expressed in the osteoclast precursor BMM and osteoclasts in mice and in monocytes, another origin of osteoclasts, in humans by searching two databases [BioGPS (http://biogps.org/#goto1/4welcome, ID:58861) and THE HUMAN PROTEIN ATLAS (https://www.proteinatlas.org/ENSG00000173198-CYSLTR1/summary/rna)] (S1A and S1B Fig). Among 87 different types of cells and tissues in mouse, BMM had the highest expression level of *Cysltr1*, while osteoclasts had the third highest expression level (S1A Fig). We also confirmed *Cysltr1* gene expression in BMM and osteoclasts from C57BL/6 mice (S2 Fig).

To clarify the role of CysLTR1 in osteoclast differentiation and mouse bone disease models, we generated *Cysltr1* mutant mice using the CRISPR-Cas9 system. We prepared two sgRNAs targeting a coding region of *Cysltr1* exon 4 to induce PCR-detectable deletions (Fig 1A). The Cas9/sgRNA complex was transferred into zygotes from BDF1 mice by electroporation (Fig 1B). Two F0 pups (#2, 4) were obtained and had *Cysltr1* gene deletion mutations, and germline transmission was successful in 2 out of 5 offspring (F1) from #4 F0 mouse (Fig 1C). DNA sequencing showed that a 112-bp deletion mutation was induced by two targeting sites and resulted in a frameshift mutation in the *Cysltr1* gene (Fig 1D and 1E, S3A Fig). We also examined off-target effects by genome editing and found no mutations in the 4 candidate off-target genes, *Mcc*, *Pde10a*, *Plcl1*, and *Ppargc1b* (S3B Fig). Therefore, we established a *Cysltr1* KO mouse line. We also generated an in-frame deletion mutant mouse (*Cysltr1^{Δ105}*) from the C57BL/6 strain, which had a 105-bp deletion in the coding region of *Cysltr1* exon 4 using the two sgRNA (S4A–S4C Fig). Deduced amino acid sequences (UniProt #Q99JA4 for reference) showed that this mutant mouse had a partially deleted CYSLT1R protein lacking the first extracellular loop and a part of the second transmembrane region (35 amino acid deletion, p. L85_Y119del). We also confirmed that there were no off-target mutations in the candidate genes (S4D Fig).

### Effects of *Cysltr1* mutations on *Cysltr1* paralog mRNA expression in bone marrow macrophages of *Cysltr1* KO and *Cysltr1^{Δ105}* mice

Recent studies revealed that genetic compensation functionally balances for target gene KO by increasing the expression of genes with high similarity to DNA sequences in some cases [21–23]. Therefore, we examined the mRNA expression of *Cysltr1* gene paralogs in BMM from *Cysltr1* KO and *Cysltr1^{Δ105}* mice. *Cysltr1* gene KO induced the complete lack of *Cysltr1* mRNA expression and stimulated the mRNA expression of *Lpar4*, but not *Cysltr2*, *Lpar6*, or *P2ry10* (S5 Fig). On the other hand, the in-frame deletion mutation (*Cysltr1^{Δ105}*) did not induce the up-regulation of paralogous gene expression (Fig 2) and, thus, appeared to avoid genetic compensation [21].

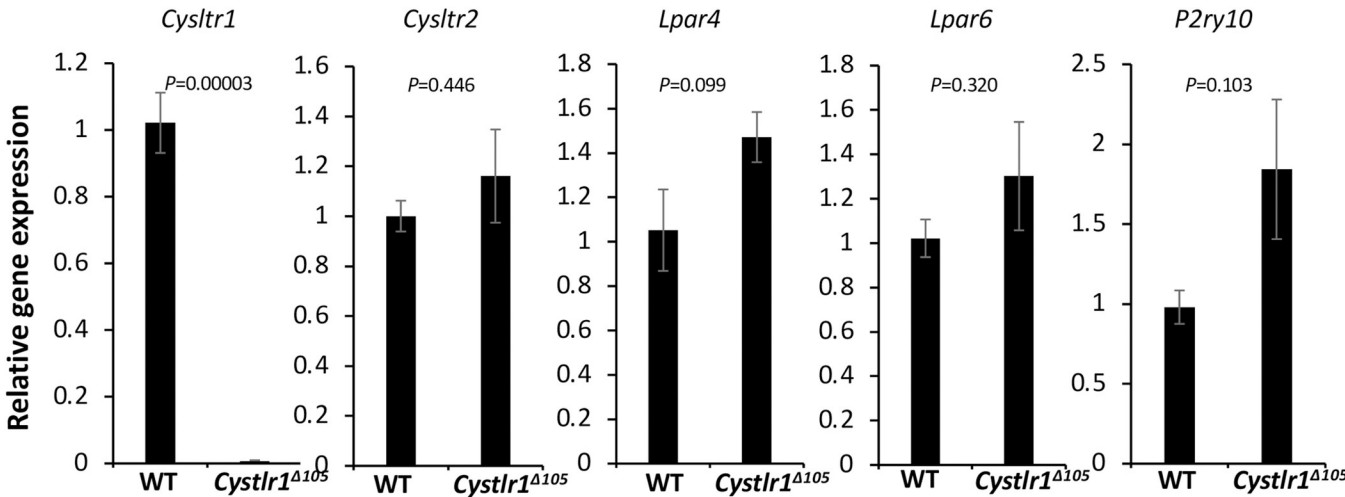

**Fig 2. Gene expression of *Cysltr1* and its paralogs in bone marrow macrophages of the *Cysltr1^{Δ105}* mouse.** A quantitative PCR analysis was performed for *Cysltr1* and its paralog mRNA expression in bone marrow macrophages from WT and *Cysltr1^{Δ105}* mice. WT, n = 4; *Cysltr1^{Δ105}*, n = 4. Gene expression was normalized with *Rn18s*.

## BMM from *Cysltr1* KO and *Cysltr1^{Δ105}* mice do not show an intracellular calcium flux in response to LTD_4

To confirm the loss of function of CYSLT1R in *Cysltr1* KO mice and *Cysltr1^{Δ105}* mice, we examined LTD_4-induced intracellular calcium flux in BMM from these mutant mice. We found that intracellular calcium was increased by ATP stimuli, but not by LTD_4 stimuli in BMM from *Cysltr1* KO or *Cysltr1^{Δ105}* mice (Fig 3A and 3B). These results indicate that the calcium response to LTD_4 was completely lost in *Cysltr1* KO and *Cysltr1^{Δ105}* mice and there was no functional genetic compensation in these *Cysltr1*-mutant mice.

## *Cysltr1* gene disruption does not suppress osteoclast formation

To clarify the role of *Cysltr1* in osteoclastogenesis *in vivo*, histological sections of the femur from WT and *Cysltr1* KO mice were analyzed using TRAP and hematoxylin staining. There was no obvious morphological change in the trabecular or cortical bones or in the number of TRAP-positive cells between *Cysltr1*-disrupted mice and WT mice (Fig 4A, S6A–S6D Fig). The CT analysis also showed that *Cysltr1* gene disruption did not induce any noticeable change in bone structural parameters of femurs under physiological conditions (S6E and S6F Fig). Furthermore, to clarify the role of *Cysltr1* in osteoclast differentiation *in vitro*, we examined whether osteoclast formation was induced by RANKL from the BMM of *Cysltr1* KO and *Cysltr1^{Δ105}* mice. However, osteoclast formation occurred normally from the BMM of *Cysltr1* KO and *Cysltr1^{Δ105}*, similar to WT BMM (Fig 4B and 4C). There was also no difference in the bone resorption activity *in vitro* between WT and *Cysltr1^{Δ105}* osteoclasts (Fig 4D). These results indicate that CysLTR1 is dispensable for osteoclast differentiation and bone resorption.

Since a previous study showed that the CysLTR1 antagonist Montelukast prevented osteoclast formation [8], we examined whether the effects of Montelukast were mediated by CysLTR1 using Cysltr1^{Δ105} BMM. Montelukast (30 μM) inhibited osteoclast formation from both WT and *Cysltr1^{Δ105}* BMM, indicating that Montelukast suppressed osteoclast formation even without a functional CysLTR1 protein (Fig 4E). In addition, we also measured size of osteoclasts. Montelukast decreased osteoclast size in WT and *Cysltr1^{Δ105}* BMMs (Fig 4F). There was no significant difference in osteoclast size between WT and *Cystlr1^{Δ105}* osteoclasts

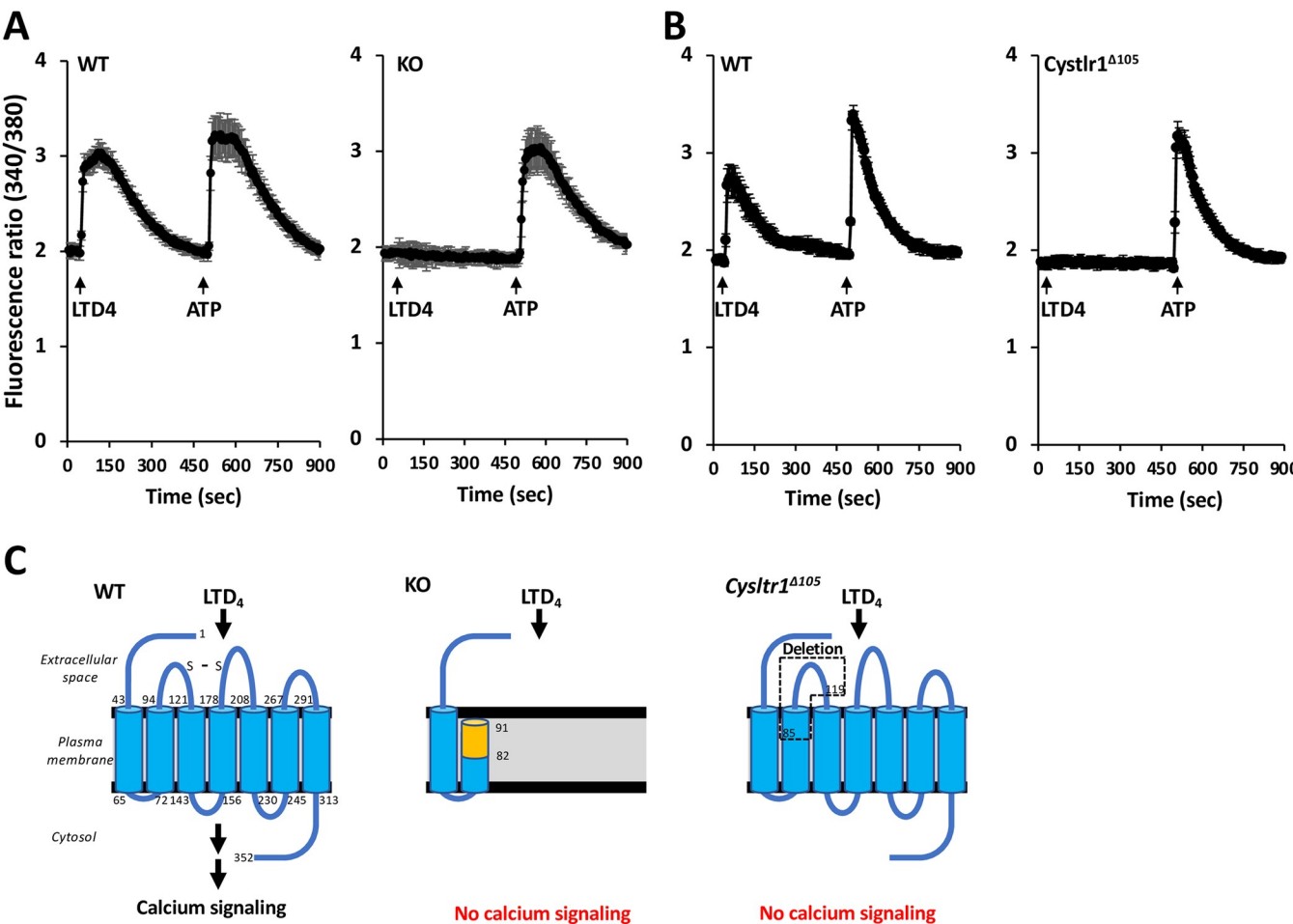

**Fig 3. Bone marrow macrophages from *Cysltr1* KO and *Cysltr1*^Δ105 mice do not show an intracellular calcium flux in response to a LTD₄ stimulation.**
BMM from WT, KO (A), and *Cysltr1*^Δ105 (B) mice were treated with Fura-2-AM and then stimulated with 1 μM leukotriene D₄ (LTD₄) and 50 μM ATP as a positive control at the indicated time points (WT n = 4, KO n = 4). Fura-2 fluorescence (340/380 nm) was detected for 900 sec (intervals of 5 sec) with a fluorescence plate reader. (C) Protein structures of WT and mutant CysLTR1. S-S indicates a disulfide bond. The yellow region of the transmembrane domain shows the unintended amino acids of KO CysLTR1. The area surrounded by the dotted line containing the first extracellular loop indicates a deleted region in CysLTR1^Δ105.

treated with Montelukast (10 μM) although we observed a tendency of suppression of Montelukast-decreased osteoclast size in *Cystlr1*^Δ105.

## LTD₄ stimulates osteoclast formation via CysLTR1 *in vitro*

A previous study showed that LTD₄ stimulated osteoclast differentiation from the macrophage cell line RAW264.7 in the presence of RANKL [12]. In addition, we previously reported that osteoclast differentiation was stimulated by GSH, which is required for the synthesis of CysLTs, including LTD₄. Therefore, we examined whether the CysLT-CysLTR1 axis exerted stimulatory effects on osteoclast differentiation. We found that LTD₄ stimulated osteoclast formation from WT BMM, but not from *Cysltr1*^Δ105 BMM on days 3 and 4 (Fig 5A), indicating that the LTD₄-CysLTR1 axis exerted stimulatory effects on osteoclast differentiation. On the other hand, GSH stimulated the formation of osteoclasts in both WT and *Cysltr1*^Δ105 BMM (Fig 5B).

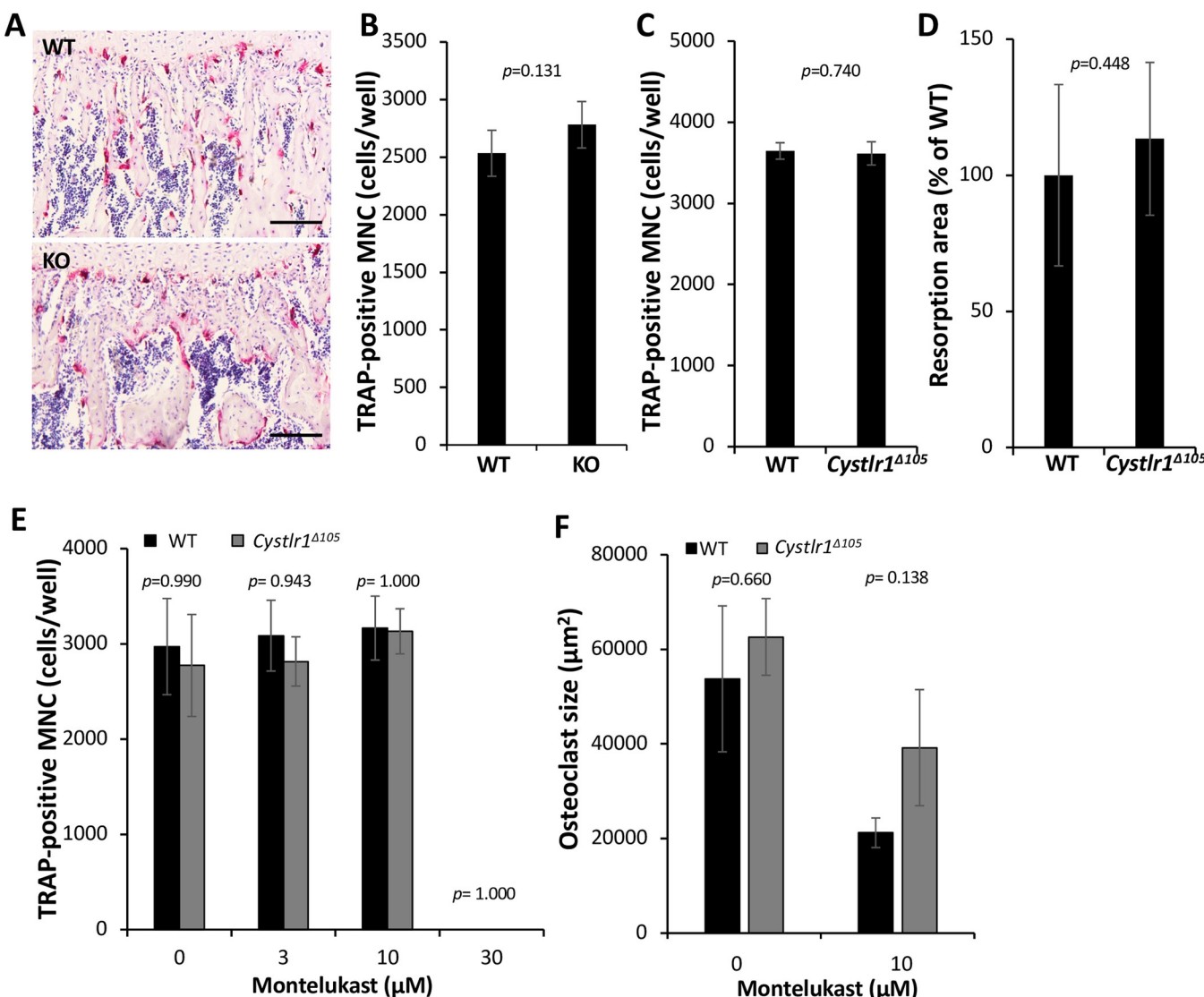

**Fig 4. Cysltr1 gene disruption does not suppress osteoclast formation and bone resorption activity.** (A) Histological sections of the femur from WT and *Cysltr1* KO mice were analyzed using TRAP and hematoxylin staining to detect osteoclasts. Representative images are shown. WT, n = 2; KO, n = 2. Scale bars: 100 μm. (B) Osteoclast formation was not affected by *Cysltr1* gene KO *in vitro*. Bone marrow macrophages (BMM) from WT and *Cysltr1* KO were cultured with 30 ng/ml RANKL for 3 days and TRAP staining was performed. WT, n = 4; *Cysltr1* KO, n = 4. (C) Osteoclast formation was not affected by *Cysltr1* gene disruption *in vitro*. BMM from WT and *Cysltr1^{Δ105}* were cultured with 50 ng/ml RANKL for 3 days and TRAP staining was performed. WT, n = 4; *Cysltr1^{Δ105}*, n = 4. (D) Effect of *Cysltr1* gene disruption on pit formation by osteoclasts *in vitro*. BMMs were treated with RANKL for 6 days on the osteoassay plate to measure bone resorption activity. WT, n = 8; *Cysltr1^{Δ105}*, n = 8. (E) The CysLTR1 antagonist Montelukast suppressed osteoclast formation from *Cysltr1^{Δ105}* BMM. BMM from WT and *Cysltr1^{Δ105}* were cultured with 100 ng/ml RANKL with the indicated concentration of Montelukast for 3 days and stained for TRAP. WT, n = 4; *Cysltr1^{Δ105}*, n = 4. (F) Montelukast decreased the size of osteoclasts from *Cysltr1^{Δ105}* BMM. Osteoclast size in (E) was measured with Image J software.

### *Cysltr1* is not involved in ovariectomy-induced osteoporosis or LPS-induced bone resorption

To clarify the role of *Cysltr1* in metabolic or inflammatory bone disease models, we performed ovariectomy in *Cysltr1* KO female mice (OVX) or injected LPS into the calvaria of *Cysltr1^{Δ105}* mice. The CT analysis showed that *Cysltr1* gene disruption did not significantly affect bone structural parameters (BV/TV, Tb. N, Tb. Th, and Tb. Sp) in femurs of the OVX model (Fig 6A and 6B) or in osteolytic areas in the LPS-induced bone resorption model (Fig 6C and 6D).

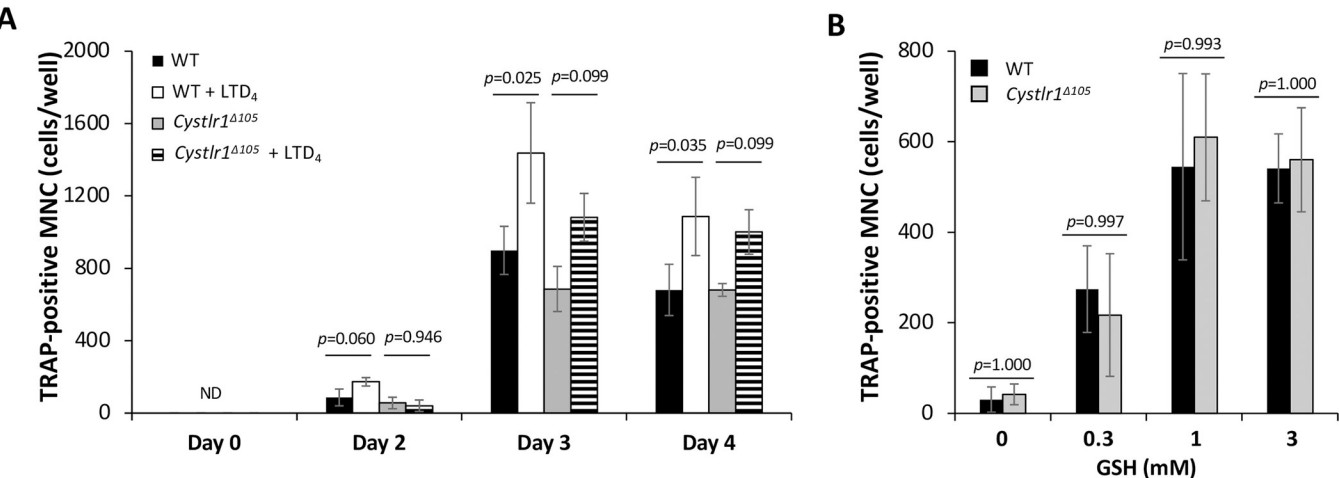

**Fig 5. Leukotriene D$_4$ stimulates osteoclast formation via functional CysLT1R *in vitro*.** (A) Leukotriene D$_4$ (LTD$_4$) stimulates osteoclast formation in WT and *Cysltr1$^{\Delta 105}$* BMM. BMM from WT and *Cysltr1$^{\Delta 105}$* were cultured with 100 ng/ml RANKL in vehicle (Ethanol) or 1 μM LTD$_4$ for 4 days. WT, n = 3; *Cysltr1$^{\Delta 105}$*, n = 3. ND: not detected. (B) Glutathione (GSH) stimulates osteoclast formation in WT and *Cysltr1$^{\Delta 105}$* BMM. BMM from WT and *Cysltr1$^{\Delta 105}$* were cultured with 10 ng/ml RANKL at the indicated concentration of GSH for 3 days. WT, n = 4; *Cysltr1$^{\Delta 105}$*, n = 4. The number of TRAP-positive cells with more than three nuclei are shown.

## Discussion

It is important to clarify the role of molecules expressed in osteoclasts in order to develop useful therapeutic strategies for diseases that cause osteopenia and bone destruction. In the present study, we generated two mutant mouse strains, *Cysltr1* KO and *Cysltr1$^{\Delta 105}$* using the CRISPR-Cas9 system. These mutants have lost calcium responses without genetic compensation. The results obtained using these mutant mice showed that CysLTR1 was dispensable for osteoclast differentiation *in vitro* and *in vivo*; however, the CysLT-CysLTR1 axis was sufficient to stimulate osteoclast differentiation. Therefore, CysLTR1 alone may be excluded as a therapeutic target for diseases that cause osteopenia and bone destruction.

A number of studies examined the relationship between leukotrienes and osteoclasts, which has remained controversial. Studies using KO mice of LTA4 hydrolase and LTC$_4$ synthase demonstrated that neutrophil-produced LTB$_4$, but not CysLTs, contributed to joint bone erosion in the rheumatoid arthritis mouse model [24], while the LTB$_4$ receptor BLT1 was involved in osteoclast activity during OVX-induced bone loss [18]. Collectively, these findings suggest that CysLTs are not involved in osteoclast differentiation or their function *in vivo*. On the other hand, analyses using RNAi knockdown, specific inhibitors (Montelukast and REV5901), and overexpression [8, 14] indicated that CysLTR1 plays a role in osteoclast differentiation, suggesting the potential involvement of CysLTR1 with ligands other than CysLTs in osteoclast differentiation. The present study using two *Cysltr1* mutant mice clearly demonstrated that CysLTR1 was dispensable for osteoclast differentiation and inflammatory bone destruction, and also that the suppression of osteoclast differentiation by Montelukast was not mediated by CysLTR1. These results showed that the CysLT-CysLTR1 axis was not essential for osteoclast differentiation or bone loss and suggested that Montelukast suppressed osteoclast differentiation via other targets, such as P2Y12 [25].

GPCRs have seven transmembrane domains, an N terminus, and three extracellular loops in the extracellular region, and three intracellular loops and a C terminus in the intracellular region. Mutations in GPCR-encoding genes have been shown to suppress their activities in most cases and cause a number of diseases [26, 27]. Various mutants in CysLTR1, a class A

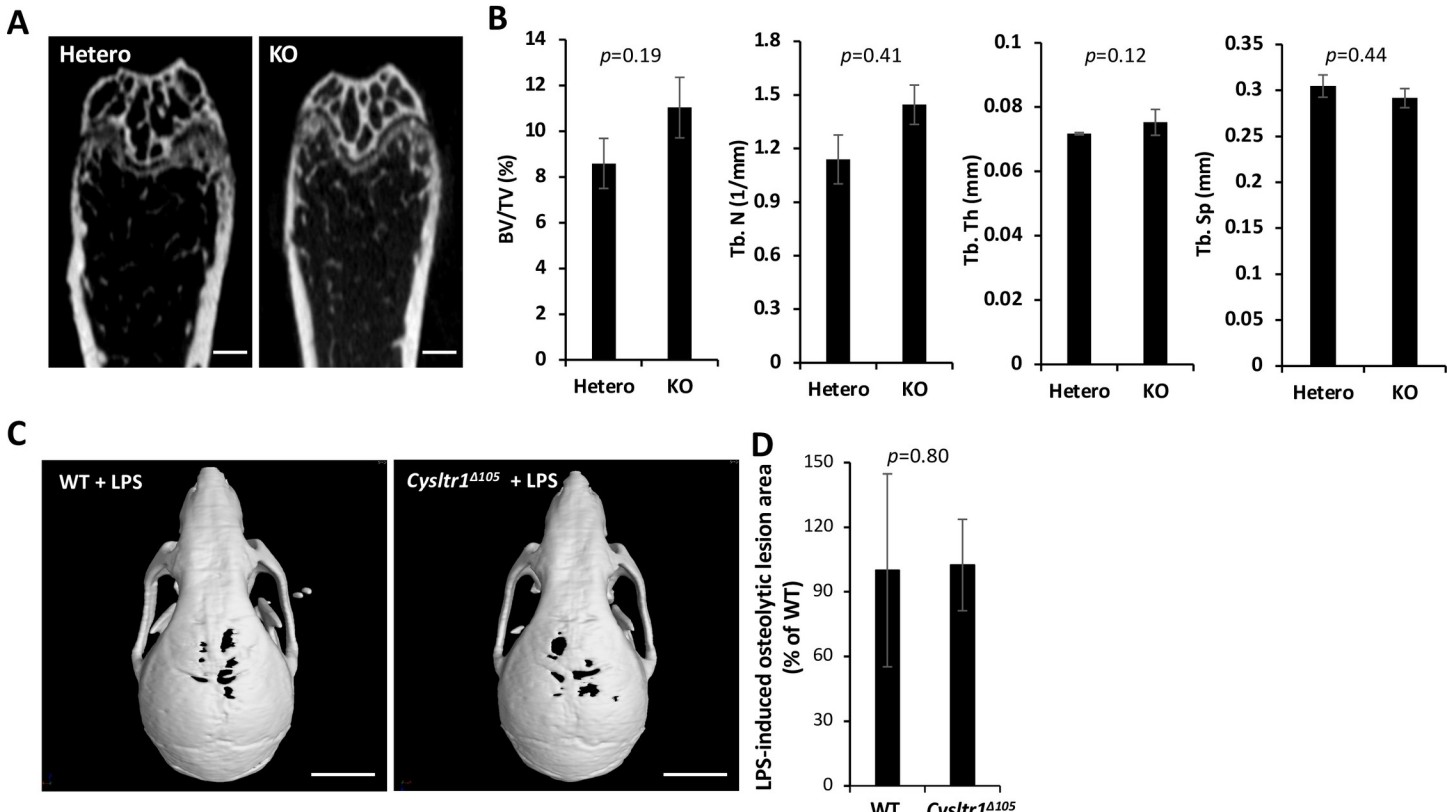

**Fig 6. *Cysltr1* gene disruption does not affect ovariectomy-induced osteoporosis or lipopolysaccharide-induced bone resorption.** (A) Micro-computed tomography (μCT) images of ovariectomy-induced osteoporosis in the femur of *Cysltr1* heterozygous and KO mice. Scale bars: 500 μm. (B) Trabecular bone volume per tissue volume (BV/TV), trabecular number (Tb.N), trabecular thickness (Tb.Th), and trabecular separation (Tb.Sp) in the metaphyseal region of the ovariectomized mouse femur. Each sample, n = 5. (C) CT images of lipopolysaccharide (LPS)-induced bone erosion in WT and *Cysltr1$^{\Delta 105}$* mice. The calvaria of WT and *Cysltr1$^{\Delta 105}$* mice treated with LPS for 5 days to induce inflammatory bone destruction were analyzed using an animal CT scanner and 3D construction software. Scale bars: 5 mm. (D) *Cysltr1$^{\Delta 105}$* did not alter the osteolytic lesion area from that in WT. Osteolytic areas were analyzed with ImageJ software. WT, n = 4; *Cysltr1$^{\Delta 105}$*, n = 5.

GPCR, have been examined: a *Cysltr1* KO mouse, in which most of exon 4, the coding region of the *Cysltr1* gene, was replaced by the *Neo* gene cassette [7], has only a portion of the N-terminal CysLTR1. The *Cysltr1* KO (frame-shift) mutant generated by Mao et al. [28] is similar to ours and has a premature stop codon within exon 4, resulting in increased vascular permeability after a zymosan stimulation. On the other hand, the L118F missense mutant, a chronic pneumonia mouse model, has one amino acid substitution in the first extracellular loop, and the sensitivity of intracellular calcium mobilization to CysLTs was found to be markedly reduced [29]. Our *Cysltr1$^{\Delta 105}$* mutant has CysLTR1 lacking the first extracellular loop, and intracellular calcium flux in response to LTD$_4$ was completely abolished. Therefore, the activity of CysLTR1 appears to be highly dependent on the first extracellular loop (Fig 3C). Since recent studies reported the involvement of *Cysltr1* in the spontaneous development of colorectal cancer and in the colitis-associated colon cancer model [30, 31], molecular target drugs for the first extracellular loop of CysLTR1 may exert therapeutic effects not only for asthma and rhinitis, but also for colon cancer.

GSH is needed to protect cells against reactive oxygen species and to synthesize CysLT, including LTD$_4$ [32]. We previously demonstrated that GSH stimulated osteoclast differentiation induced by RANKL or RANKL along with TNFα. In the present study, GSH stimulated RANKL-induced osteoclast differentiation from *Cysltr1$^{\Delta 105}$* as well as WT BMM. These results

suggest that the stimulatory effects of GSH were not dependent on the function of CysLTR1 and also that the target of GSH was not the CysLT-CysLTR1 axis.

Therefore, we concluded that CysLTR1 is not involved in osteoclast differentiation *in vitro* or bone loss *in vivo* and that the first extracellular loop of CysLTR1 has potential as a useful drug target, such as an anti-CysLTR1 drug, via the prevention of calcium signaling.

## Supporting information

**S1 Fig. *Cysltr1* is expressed at higher levels in the osteoclast lineage than in other cells and tissues.** A list of the top 25 *Cysltr1*-expressing cells and tissues in mice (A) and humans (B). These data were obtained from the public databases BIOGPS and THE HUMAN PROTEIN ATLAS.
(PDF)

**S2 Fig. Effects of osteoclast differentiation from bone marrow macrophages on *Cysltr1* and its paralog mRNA expression.** Bone marrow macrophages (BMM) were cultured with RANKL for 3 days for osteoclast (OC) formation. A qPCR analysis of *Cysltr1* and its paralog mRNA expression levels in WT (C57BL/6) BMM and OC. BMM, n = 4; OC, n = 4. Gene expression was normalized with *18s ribosomal RNA* (*Rn18s)*.
(PDF)

**S3 Fig. Deduced amino acid sequences of CysLTR1 and DNA sequences of off-target candidate genes for *Cysltr1* knockout (KO) mice.** (A) Amino acid sequences deduced from the DNA sequences shown in Fig 1D. (B) Off-targeting candidate genes, which have a similar sequence to 12 nucleotides plus PAM targeted for the *Cysltr1* gene. No off-target effects were found in the *Cysltr1* KO mouse genome.
(PDF)

**S4 Fig. Generation of the in-frame deletion mutant mouse *Cysltr1$^{\Delta105}$* by the CRISPR-Cas9 system.** (A) Genotyping PCR of F0 (left) and F1 (right) pups after *Cysltr1* gene editing in C57BL/6. The amplicon size for the wild-type (WT) *Cysltr1* gene was 818 bp and that for the mutated gene was 713 bp. The #4 F0 female mouse was mated to a WT stud, providing the *Cysltr1$^{\Delta105}$* strain (F1 mice). Sequencing chromatogram (B) and DNA sequence (C) of the mutation site in *Cysltr1$^{\Delta105}$*. The deletion size of the mutation site was 105 bp. (D) Deduced amino acid sequences of CysLTR1. (E) Sequencing chromatograms of the off-targeting candidate genes with 12-mer of the *Cysltr1* target sequences adjacent to the PAM. No off-targeting was found in the *Cysltr1* mutant mouse genome. "-" indicates deleted nucleotide bases.
(PDF)

**S5 Fig. Gene expression of *Cysltr1* and its paralogs in bone marrow macrophages from the *Cysltr1* KO mouse.** A quantitative PCR analysis was performed for *Cysltr1* and its paralog mRNA expression in bone marrow macrophages from WT and *Cysltr1* KO mice. WT, n = 3; KO, n = 3. Gene expression was normalized with *glyceraldehyde-3-phosphate dehydrogenase* (*Gapdh)* as a housekeeping gene.
(PDF)

**S6 Fig. *Cysltr1* gene disruption does not affect bone structure of femur.** Histomorphometry of the femur from WT, *Cysltr1* KO (A and B) or *Cysltr1$^{\Delta105}$* (C and D) were analyzed using TRAP and hematoxylin staining to detect osteoclasts. Each sample, n = 2. (E) Micro-computed tomography (μCT) images of the femur of WT and *Cysltr1$^{\Delta105}$* mice under physiological condition. Scale bar: 500 μm. (F) Trabecular bone volume per tissue volume (BV/TV), trabecular number (Tb.N), trabecular thickness (Tb.Th), and trabecular separation (Tb.Sp) in the

metaphyseal region of the femur of (E). Each sample, n = 2.
(PDF)

**S1 Raw images. Raw agarose gel images of Fig 4 and S4 Fig.**
(PDF)

**S1 Data.**
(XLSX)

**S1 Table.**
(PDF)

# Acknowledgments

We thank Drs Tetsuya Bando, Hirotsugu Kobuchi, and Keita Sato for their fruitful discussions.

# Author Contributions

**Funding acquisition:** Hirofumi Fujita, Hideyo Ohuchi.

**Investigation:** Hirofumi Fujita, Aoi Ando, Mitsuaki Ono, Toshitaka Oohashi.

**Methodology:** Hirofumi Fujita, Takako Hattori, Munenori Habuta, Satoshi Kubota, Hideyo Ohuchi.

**Project administration:** Hirofumi Fujita, Hideyo Ohuchi.

**Resources:** Yohei Mizusawa.

**Supervision:** Hideyo Ohuchi.

**Validation:** Hirofumi Fujita.

**Writing – original draft:** Hirofumi Fujita, Hideyo Ohuchi.

**Writing – review & editing:** Hirofumi Fujita, Toshitaka Oohashi, Satoshi Kubota, Hideyo Ohuchi.

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
