## [Decision Letter · Decision Letter 0]

25 Aug 2022

PONE-D-22-18067Cysteinyl leukotriene receptor 1 is dispensable for osteoclast differentiation and bone resorptionPLOS ONE

Dear Dr. Fujita,

Thank you for submitting your manuscript to PLOS ONE. After careful consideration, we feel that it has merit but does not fully meet PLOS ONE’s publication criteria as it currently stands. Therefore, we invite you to submit a revised version of the manuscript that addresses the points raised during the review process.

We look forward to receiving your revised manuscript.

Kind regards,

Jung-Eun Kim

Academic Editor

PLOS ONE

Journal Requirements:

"Grant numbers awarded to each author

Hirofumi Fujita: JSPS KAKENHI 15K10475, 19K09625

The full name of each funder

JSPS: Japan Society for the Promotion of Science

Promotion of Science and Technology in Okayama Prefecture by the Ministry of Education, Culture, Sports, Science and Technology of Japan.

URL of each funder website

https://www.jsps.go.jp/english/index.html

Did the sponsors or funders play any role in the study design, data collection and analysis, decision to publish, or preparation of the manuscript?: NO"

Reviewers' comments:

Reviewer's Responses to Questions

**Comments to the Author**

1. Is the manuscript technically sound, and do the data support the conclusions?

Reviewer #1: Yes

Reviewer #2: Yes

Reviewer #3: Yes

2. Has the statistical analysis been performed appropriately and rigorously? 

Reviewer #1: Yes

Reviewer #2: Yes

Reviewer #3: Yes

3. Have the authors made all data underlying the findings in their manuscript fully available?

Reviewer #1: Yes

Reviewer #2: Yes

Reviewer #3: Yes

4. Is the manuscript presented in an intelligible fashion and written in standard English?

Reviewer #1: Yes

Reviewer #2: Yes

Reviewer #3: Yes

5. Review Comments to the Author

Reviewer #1: In the study by Fujita et al. the authors create a novel global mouse knockout model of the cysteinyl leukotriene receptor 1 (CysTR1). The authors analyze the skeletal phenotype of these mice. Unfortunately, the authors demonstrate that CyslTR1 is not necessary for osteoclast differentiation. The loss of CysTR1 does not lead to changes in the skeleton under pathological conditions. The authors do demonstrate successfully loss of Cysltr1 expression by recombination and that other related genes are not changed in expression due to the knockout of Cysltr1. There are some minor concerns that should be addressed and are listed below.

1. In Figure 4 the images in panel A should be redone as there is no contrast and it is difficult to see the TRAP positive cells.

2. Besides counting number in panel 4D the authors should also count size of the osteoclasts.

3. Should be stated in the text of the manuscript why the authors chose not to analyze the skeleton under physiological conditions.

Reviewer #2: This paper is an interesting analysis of the role of cysteinyl leukotriene receptor 1 in osteoclastic bone resorption. Experimental results using cell culture systems and bone resorption-promoting mouse models clearly demonstrate that cysteinyl leukotriene receptor 1 is not directly involved in osteoclast differentiation and bone resorption.

Reviewer #3: The authors investigated roles of cysteinyl leukotriene receptor 1 (CysLTR1) in osteoclastogenesis. To clarify the role in vivo they generated two lines of CysLTR1 mutant mice. Macrophages from those mutant mice did not respond to the stimulation of LTD4. However, these mutants failed to impact osteoclastogenesis in in vivo and in vitro studies. The experiments seems to be well organized and performed, and their conclusion is based on solid data. However, there are some concerns in the present manuscript.

Major points

1. Figure 4A…To clearly show no difference in osteoclast number between WT and KO, the quantitative data of osteoclast number are needed. In addition, scale bar is missing.

2. The authors only focused on the roles of CysLTR1 in osteoclast differentiation. To confirm whether the deficiency of CysLTR1 impacts on osteoclastic bone resorbing activity, serum bone resorption parameter such as serum CTX should be analyzed.

3. The explanation of roles of CysLT in bone resorption seems to be too short in Introduction. The authors should explain why they focus on CysLT in bone metabolism and roles of CysLT in bone metabolism and bone diseases in more detail.

4. Fig. 6…Scale bars are missing in panels A and D.

6. PLOS authors have the option to publish the peer review history of their article (what does this mean?). If published, this will include your full peer review and any attached files.

Reviewer #1: No

Reviewer #2: No

Reviewer #3: No

---

## [Author Response · Author response to Decision Letter 0]

7 Oct 2022

Answers to reviewer’s comments

Reviewer #1: 

General comment

In the study by Fujita et al. the authors create a novel global mouse knockout model of the cysteinyl leukotriene receptor 1 (CysTR1). The authors analyze the skeletal phenotype of these mice. Unfortunately, the authors demonstrate that CyslTR1 is not necessary for osteoclast differentiation. The loss of CysTR1 does not lead to changes in the skeleton under pathological conditions. The authors do demonstrate successfully loss of Cysltr1 expression by recombination and that other related genes are not changed in expression due to the knockout of Cysltr1. There are some minor concerns that should be addressed and are listed below.

Comment 1. 

In Figure 4 the images in panel A should be redone as there is no contrast and it is difficult to see the TRAP positive cells.

Answer to comment 1

We thank the reviewer #1 for the kind remarks. According to your advice, we replaced the images to high magnification images in Fig 4A of the Revised Manuscript with Track Changes.

Comment 2. 

Besides counting number in panel 4D the authors should also count size of the osteoclasts.

Answer to comment 2 

According to your advice, we measured size of osteoclasts using ImageJ software. The data showed that Montelukast decreased osteoclast size in WT and Cystlr1Δ105 BMMs. There was no significance in osteoclast size between WT and Cystlr1Δ105 osteoclasts treated with Montelukast although we observed a tendency of suppression of Montelukast-decreased osteoclast size in Cystlr1Δ105. We added this results and sentences to Revised Manuscript with Track Changes; the Fig 4F; the Results section line 423-427 and line 445-446. 

Comment 3. 

Should be stated in the text of the manuscript why the authors chose not to analyze the skeleton under physiological conditions.

Answer to Comment 3. 

In response to the suggestion, we added the data of bone morphological analysis under physiological conditions. The results of histomorphometry and CT analysis did not show any noticeable differences between WT and Cystlr1 mutants. We added these results and sentences to Revised Manuscript with Track Changes; the S6 Fig; the Results section line 406 and 409; the Supporting information section line 686-693.

 

Reviewer #2: 

General comment 

This paper is an interesting analysis of the role of cysteinyl leukotriene receptor 1 in osteoclastic bone resorption. Experimental results using cell culture systems and bone resorption-promoting mouse models clearly demonstrate that cysteinyl leukotriene receptor 1 is not directly involved in osteoclast differentiation and bone resorption.

Answer to comment

We thank the peer review of reviewer #2. 

 

Reviewer #3:

General comment

The authors investigated roles of cysteinyl leukotriene receptor 1 (CysLTR1) in osteoclastogenesis. To clarify the role in vivo they generated two lines of CysLTR1 mutant mice. Macrophages from those mutant mice did not respond to the stimulation of LTD4. However, these mutants failed to impact osteoclastogenesis in in vivo and in vitro studies. The experiments seems to be well organized and performed, and their conclusion is based on solid data. However, there are some concerns in the present manuscript.

Major points

Comment 1. 

Figure 4A…To clearly show no difference in osteoclast number between WT and KO, the quantitative data of osteoclast number are needed. In addition, scale bar is missing.

Answer to Comment 1. 

We thank the reviewer #3 for the kind remarks. According to your advice, we added quantitative data of osteoclast number of WT, KO and Cysltr1Δ105 femur although the sample size was not sufficient due to dead line of revised manuscript submission. The results of histomorphometry analysis did not show any noticeable differences between WT and Cystlr1 mutants. We added scale bar in Fig 4A. We added these results and sentences to the Revised Manuscript with Track Changes; the S6 Fig A-D; the Materials and methods section line 204-206; the Results section line 432; the Supporting information section line 686-689.　 

Comment 2. 

The authors only focused on the roles of CysLTR1 in osteoclast differentiation. To confirm whether the deficiency of CysLTR1 impacts on osteoclastic bone resorbing activity, serum bone resorption parameter such as serum CTX should be analyzed.

Answer to Comment 2. 

As you pointed out, our study lacked an analysis of the bone resorbing activity of osteoclasts. We performed pit formation assay for bone resorbing activity of osteoclast differentiated from WT and Cysltr1Δ105 BMMs from using osteoassay plate. The data showed that there was no difference in the bone resorption activity in vitro between WT and Cysltr1Δ105 osteoclasts. We added this result and sentences to the Revised Manuscript with Track Changes; the Fig 4D; the Materials and methods section line 213-221; the Results section line 413-417, line 429-430 and line 439-441.　

Comment 3. 

The explanation of roles of CysLT in bone resorption seems to be too short in Introduction. The authors should explain why they focus on CysLT in bone metabolism and roles of CysLT in bone metabolism and bone diseases in more detail.

Answer to Comment 3. 

Thank you for your valuable suggestions. According to your advice, we added the explanation of CysLTs role in osteoclast differentiation and bone resorption. It explained that CysLTs stimulate isolated osteoclasts to form resorption pits on calcified matrices and stimulate bone resorption in organ culture of mouse calvariae. In addition, LTD4 stimulates osteoclast differentiation from macrophage cell line RAW264.7 in the presence of RANKL. Previous our study showed that the CysLT synthesis substrate, GSH stimulated osteoclast differentiation. We added these sentences to the Revised Manuscript with Track Changes; the Introduction section line 71-77 and line 80-81. We removed sentences from next paragraph line 84.

Comment 4. 

Fig. 6…Scale bars are missing in panels A and D.

Answer to Comment 4. 

We added scale bar in Fig 6A and D. We added a sentence to the Revised Manuscript with Track Changes; the Results section line 481 and line 487.

---

## [Decision Letter · Decision Letter 1]

24 Oct 2022

Cysteinyl leukotriene receptor 1 is dispensable for osteoclast differentiation and bone resorption

PONE-D-22-18067R1

Dear Dr. Fujita,

We’re pleased to inform you that your manuscript has been judged scientifically suitable for publication and will be formally accepted for publication once it meets all outstanding technical requirements.

Kind regards,

Jung-Eun Kim

Academic Editor

PLOS ONE

Additional Editor Comments (optional):

Reviewers' comments:

Reviewer's Responses to Questions

**Comments to the Author**

1. If the authors have adequately addressed your comments raised in a previous round of review and you feel that this manuscript is now acceptable for publication, you may indicate that here to bypass the “Comments to the Author” section, enter your conflict of interest statement in the “Confidential to Editor” section, and submit your "Accept" recommendation.

Reviewer #1: All comments have been addressed

Reviewer #3: All comments have been addressed

2. Is the manuscript technically sound, and do the data support the conclusions?

Reviewer #1: Yes

Reviewer #3: Yes

3. Has the statistical analysis been performed appropriately and rigorously? 

Reviewer #1: Yes

Reviewer #3: (No Response)

4. Have the authors made all data underlying the findings in their manuscript fully available?

Reviewer #1: Yes

Reviewer #3: Yes

5. Is the manuscript presented in an intelligible fashion and written in standard English?

Reviewer #1: Yes

Reviewer #3: Yes

6. Review Comments to the Author

Reviewer #1: (No Response)

Reviewer #3: The authors have addressed concerns from this reviewer. The manuscript is largely improved. This reviewer has no concerns anymore.

7. PLOS authors have the option to publish the peer review history of their article (what does this mean?). If published, this will include your full peer review and any attached files.

Reviewer #1: No

Reviewer #3: No

---

## [Editor Report · Acceptance letter]

7 Nov 2022

PONE-D-22-18067R1 

Cysteinyl leukotriene receptor 1 is dispensable for osteoclast differentiation and bone resorption 

Dear Dr. Fujita:

I'm pleased to inform you that your manuscript has been deemed suitable for publication in PLOS ONE. Congratulations! Your manuscript is now with our production department. 

Kind regards, 

on behalf of

Dr Jung-Eun Kim 

Academic Editor

PLOS ONE